Cloning, expression and characterization of a cold-adapted endo-1, 4-β-glucanase from Citrobacter farmeri A1, a symbiotic bacterium of Reticulitermes labralis

Bai Xi 1
Yuan Xianjun 1
Wen Aiyou 2
Li Junfeng 1
Bai Yunfeng 3
Shao Tao taoshaolan@163.com 1
1 Institute of Ensiling and Processing of Grass, Nanjing Agricultural University , Nanjing , China
2 College of Animal Science, University of Science and Technology of Anhui , Fengyang , China
3 Institute of Agricultural Resource and Environment, Jiangsu Academy of Agricultural Sciences , Nanjing , China
Gatti-Lafranconi Pietro
Electronic publication date: 2016 Nov 8
Publication date: 2016
Volume: 4
Electronic Location ID: e2679
Received 2016 Jun 22; Accepted 2016 Oct 12
Copyright: ©2016 Bai et al.
Copyright year: 2016
Copyright holder: Bai et al.
License: This is an open access article distributed under the terms of the Creative Commons Attribution License, which permits unrestricted use, distribution, reproduction and adaptation in any medium and for any purpose provided that it is properly attributed. For attribution, the original author(s), title, publication source (PeerJ) and either DOI or URL of the article must be cited.
License URL: https://creativecommons.org/licenses/by/4.0/

Keywords: Citrobacter farmeri, Endoglucanase, Cold-adapted, Expression, Escherichia coli, Properties

Funding: Independent Innovation of Agricultural Sciences CX(15)1003 Network and Technology Served of Chinese Academy of Sciences Program (STS) KFJ-EW-STS-071 National Spark Plan Project 2013GA840003 This work was supported by Independent Innovation of Agricultural Sciences in Jiang Su Province [CX(15)1003], Network and Technology Served of Chinese Academy of Sciences Program (STS) “Grassland agricultural system construction and industrialization demonstration of typical village (Jina village) in Tibet” (KFJ-EW-STS-071), and the National Spark Plan Project “Integration and demonstration of planting high quality forage grass and breeding cows healthily technology” (2013GA840003). The funders had no role in study design, data collection and analysis, decision to publish, or preparation of the manuscript.

==============================
Background

Many biotechnological and industrial applications can benefit from cold-adapted EglCs through increased efficiency of catalytic processes at low temperature. In our previous study, Citrobacter farmeri A1 which was isolated from a wood-inhabiting termite Reticulitermes labralis could secrete a cold-adapted EglC. However, its EglC was difficult to purify for enzymatic properties detection because of its low activity (0.8 U/ml). The objective of the present study was to clone and express the C. farmeri EglC gene in Escherichia coli to improve production level and determine the enzymatic properties of the recombinant enzyme.

Methods

The EglC gene was cloned from C. farmeri A1 by thermal asymmetric interlaced PCR. EglC was transformed into vector pET22b and functionally expressed in E. coli. The recombination protein EglC22b was purified for properties detection.

Results

SDS-PAGE revealed that the molecular mass of the recombinant endoglucanase was approximately 42 kDa. The activity of the E. coli pET22b-EglC crude extract was 9.5 U/ml. Additionally, it was active at pH 6.5–8.0 with an optimum pH of 7.0. The recombinant enzyme had an optimal temperature of 30–40 °C and exhibited >50% relative activity even at 5 °C, whereas it lost approximately 90% of its activity after incubation at 60 °C for 30 min. Its activity was enhanced by Co2+ and Fe3+, but inhibited by Cd2+, Zn2+, Li+, Triton X-100, DMSO, acetonitrile, Tween 80, SDS, and EDTA.

Conclusion

These biochemical properties indicate that the recombinant enzyme is a cold-adapted endoglucanase that can be used for various industrial applications.

Introduction

Cellulases, secreted by bacteria, fungi, and invertebrates, degrade β-1,4 glycosidic bonds of cellulose into sugars (Ogura et al., 2006; Limayem & Ricke, 2012; Shelomi et al., 2014). The complete degradation of cellulose requires three different cellulases: endo-1,4-β-glucanase, exo-1,4-β-glucanase, and 1,4-β–glucosidase (Ozioko, Eze & Chilaka, 2013). Endo-β-1,4-glucanase (EglC, EC 3.2.1.4) is an important enzyme that hydrolyzes glycosidic linkages and releases oligosaccharides of different lengths. Some studies have suggested that EglC is more effective than exoglucanase and glucosidase for degrading cellulose (Yasir et al., 2013).

Numerous biotechnological and industrial applications can benefit from cold-adapted EglCs through increased efficiency of catalytic processes at low temperature (Kasana & Gulati, 2011). However, cold-adapted EglCs are generally not thermally stability and can easily be inactivated by elevated temperatures (Vester, Glaring & Stougaard, 2014) Such processes have the merits of saving production and energy costs, maintaining taste and other organoleptic characteristics and reducing the risk of contamination (Vester, Glaring & Stougaard, 2014).

For application in the biofuel industry, cold-adapted EglCs can produce ethanol from cellulosic material at low temperature (Cavicchioli et al., 2011). However, the biological conversion of cellulose to bioethanol is typically performed at relatively high temperatures (50–60 °C), which can increase energy consumption and production costs (Cavicchioli et al., 2011; Tiwari et al., 2015) Cold-adapted EglCs can also be used in the degradation of polymer in pulp and paper processes, stonewashing and biopolishing of textiles, and in the food, silage and feed industries (Kasana & Gulati, 2011). Based on these applications, cold-adapted EglCs have attracted increased attention (Maharana & Ray, 2015; Gerday et al., 2000). However, few cold-adapted β-glucanases have been identified and cloned to date (Bhat et al., 2013; Cavicchioli et al., 2011)

EglC genes from some microorganisms have been cloned and expressed in Escherichia coli for secretion of endoglucanases (Kasana & Gulati, 2011). Termites have a rich variety of commensal microbes in their intestinal tracts that efficiently digest lignocellulose, and are thus considered as promising reservoirs of microbial symbionts and enzymes with high biotechnological potential (Brune, 2014). In our previous study, Citrobacter farmeri A1 that secreted cold-adapted EglC was isolated from the wood-inhabiting termite Reticulitermes labralis (X Bai & T Shao, 2015, unpublished data). However the activity of this EglC was low (0.8 U/ml). The objective of the present work was to clone and express the EglC gene of C. farmeri to improve production level and determine the enzymatic properties of the recombinant enzyme.

Materials and Methods

Plasmids, chemicals, and culture medium

C. farmeri A1 that secreted cold-adapted EglC was isolated from R. labralis and stored in our laboratory. The plasmid pMD20-T vector (TaKaRa, Dalian, China) and E. coli DH5α were used for gene cloning, whereas the pET22b vector (Novagen, Madison, WI) and E. coli BL21 (laboratory stock) were used for gene expression. The flanking regions of the EglC gene were amplified using the TaKaRa Genome Walking Kit (TaKaRa, Dalian, China). Restriction endonucleases, T4 DNA ligase, and DNA polymerase were purchased from TaKaRa (Dalian, China). TIANgel Midi Purification Kit, TIANprep Mini Plasmid Kit, and 2 × Taq PCR Master Mix were purchased from Tiangen (Beijing, China).

EglC gene cloning

Genomic DNA of C farmeri A1 was isolated using the TIANamp Bacteria DNA Kit (Tiangen, Beijing, China). The core region of the EglC gene was amplified using the degenerate primers GH8-F and GH8-R (Table 1), which were designed based on two conserved blocks, EGQSY[A/G][M/L]FFAL and DAIRVY[L/M]WAG[M/L], of the glycoside hydrolase family 8 (GH8) bacterial endoglucanases. Touchdown PCR conditions were as follows: an initial denaturation step of 5 min at 95°C and 15 cycles of 30 s at 94 °C, 30 s at 70 °C, and 20 s at 72 °C, followed by 30 cycles of 30 s at 94 °C, 30 s at 55 °C, and 40 s at 72 °C, and a final elongation step of 10 min at 72 °C. Amplified DNAs of the appropriate size were purified and ligated into the pMD20-T vector, and confirmed by DNA sequencing (Genscript Corporation, Nanjing, China).

Table 1 Primers used in this study.

Primer name	Primer sequence (5′→ 3′)	
GH8-F	GARGGNCARWSNTAYGCNATGTTYTTYGC	
GH8-R	CATNCCNGCCCANARRTANACNCKDAT	
FSP1	TCTCTGGCTTCAGTTGCCAGCCTT	
FSP2	AAGCGGGCGAAATACTGCGCCAGCT	
FSP3	GCACCGTTACCACTTCTTCGCT	
C1	GTNCGASWCANAWGTT	
RSP1	TGGTTTTGCCGAAGCCAACGC	
RSP2	TGCGCGAAACCAACCAGCGACT	
RSP3	CCAGAGAAAACGCTGGTCAG	
C2	NGTCGASWGANAWGAA	
GHF	ATGAACGCGTTGCGTAGTGG	
GHR	TTAATTTGAACTTGCGCATTCCT	
EglC-F	CTTCCATGGGCCTGTACCTGGCCCGCAT (NcoI)	
EglC-R	CCGCTCGAGATTTGAACTTGCGCATTCCTGG (XhoI)	

Full-length EglC was amplified by thermal asymmetric interlaced PCR (TAIL-PCR) using nested insertion-specific primers designed based on the conserved domain. The 5′-flanking genomic sequences were amplified using FP-specific primers (FSP1, FSP2, and FSP3; Table 1) and the random degenerate primer C1, whereas the 3′-flanking genomic sequences were amplified with RP-specific primers (RSP1, RSP2, and RSP3; Table 1) and the random degenerate primer C2 using the TaKaRa Genome Walking Kit (TaKaRa, Dalian, China). The TAIL-PCR products were ligated into the pMD20-T vector and transformed into E. coli DH5α for DNA sequencing. Then, the 5′- and 3′-flanking regions were assembled with the core sequence.

Based on the assembled sequence, the GHF and GHR primers (Table 1) were used to amplify the full length of the EglC gene. The PCR program included an initial step of 6 min at 95 °C, a second step of 35 cycles including 35 s at 95 °C, 1 min at 60 °C and 1 min at 72 °C, and a final step of 8 min at 72 °C The DNA product was cloned into the pMD20-T vector using standard procedures and sequenced by Genscript Corporation (China).

Construction and transformation of the recombinant expression vectors

For expression of EglC in E. coli, the mature protein-coding sequences (signal peptide excluded) were amplified by PCR from genomic DNA of C. farmeri A1. The PCR primers were EglC-F and EglC-R containing NcoI and XhoI sites (Table 1). The PCR program included an initial step of 6 min at 95 °C, a second step of 35 cycles including 35 s at 95 °C, 1 min at 58 °C and 1 min at 72 °C, and a final step of 8 min at 72 °C The purified PCR products were ligated into the pMD20-T plasmid. The vector pMD20-EglC was digested with NcoI and XhoI, and then the purified EglC gene was cloned to the pET22b vector. The resulting plasmid pET22b-EglC, was transformed into E. coli BL21 and plated on LB agar containing 100 µg ml−1 ampicillin to select positive transformants. The positive clones were checked by colony PCR using EglC-F and EglC-R primers (Table 1). The plasmid pET22b-EglC was sequenced by Genscript Corporation (China).

Expression of the EglC gene in E. coli and purification of the recombinant protein

The recombinant E. coli pET22b-EglC was cultured at 37 °C in LB medium with ampicillin. When the culture density reached an optical density of approximately 0.5 at 600 nm, isopropyl-b-D-1-thiogalactopyranoside (IPTG) was added into the medium to a final concentration of 1 mM to induce endoglucanase expression. After 4 h, cultured cells were collected by centrifugation at 10,000 ×g for 6 min at 4 °C, resuspended in ice-cold buffer (Na2HPO4-citric acid; pH 7), and disrupted by sonication. Proteins were obtained by centrifugation and purified using a Ni2+ affinity chromatography column (CWBIO, China). Expressed proteins were detected by sodium dodecyl sulfate-polyacrylamide gel electrophoresis (SDS-PAGE), and their concentration was determined using a Micro BCA protein assay kit (Jiancheng, Nanjing, China).

Detection of enzymatic activities

The activity of Eglc22b was determined using the 3,5-dinitrosalicylic (DNS) method as described by Fu et al. (2010). Sodium-carboxymethyl cellulose (CMC-Na) was used as a substrate at a concentration of 1.0% (w/v). The standard reaction mixture, containing 1 ml of appropriately diluted enzyme and 1 ml of CMC-Na buffer (pH 7), was incubated at 40 °C for 30 min. The reaction was terminated by addition of 3 ml of DNS to the mixture and 5 min of boiling. The release of reducing sugars was measured at 540 nm by using a spectrophotometer (Shanghai Precision & Scientific Instrument Co., Shanghai, China). One glucanase unit was defined as the amount of enzyme required to release 1 µmol of glucose per minute at the assay temperature.

Biochemical characteristics of EglC22b

The optimum pH for the activity of EglC22b was determined at 40 °C for 30 min in Na2HPO4-citric acid buffer (pH 3.5–7) and sodium phosphate buffer (pH 8.0). To determine the pH stability, EglC22b was incubated at 40 °C for 30 min in different buffers of pH 2.5 to 6.5 (1.0 interval), and the residual enzyme activity was measured at optimal pH and temperature using the DNS method as described previously.

The optimal temperature for the activity of EglC22b was determined in Na2HPO4-citric acid buffer (pH 7.0) at 10 °C to 70 °C (10 °C interval) for 30 min. To determine the thermal stability, EglC22b was incubated at 20, 30, 40, 50, and 60 °C for 30 min each, and the residual enzyme activity was measured at optimal pH and temperature using the DNS method as described previously.

The effects of different metal ions and chemical reagents on the activity of EglC22b were assessed by incubating the enzyme at 40 °C for 30 min in a standard reaction mixture of Na2HPO4-citric acid buffer (pH 7.0) containing CoCl2 (2 mM), CdCl2 (2 mM), ZnCl2 (2 mM), FeCl3 (2 mM), Li2SO4 (2 mM), Triton X-100 (10%), dimethyl sulfoxide (DMSO, 20%), acetonitrile (20%), Tween 80 (20%), SDS (0.4%), and ethylene-diamine-tetraacetic acid (EDTA, 20 mM). An assay system without the added metal ions was used as a control, and the enzyme activity was measured using the DNS method as described previously.

Statistical analysis

All measurements of the present study were carried out in duplicate. Data were analyzed by Microsoft Excel 2010. Data were presented as means with standard deviation (SD).

Results

Cloning and sequencing of EglC

An approximately 600-bp core region of EglC was amplified by touchdown PCR from C. farmeri A1. Based on the sequence of the EglC core region, TAIL-PCR was used to amplify the flanking sequences of the DNA fragment. Finally, a 1,107 bp fragment was obtained from C farmeri A1 (Fig. 1). The open reading frame was predicted to encode a protein of 368 amino acids with a theoretical molecular mass of 39.1 kDa. Sequence analysis revealed a signal peptide with a length of 21 amino acids at the N-terminal of the enzyme. A BLAST search was used to predict the mature EglC of C. farmeri, revealing that the mature EglC belongs to glycoside hydrolase family 8 (Fig. 2). Additionally, the deduced EglC had 86.4%, 43.4%, 44.7%, 39.7%, and 44.3% amino acid sequence identity with endo-β-1,4-glucanases from E. coli CFT073, Burkholderia sp. CCGE1002, Cupriavidus taiwanensis, Pseudomonas fluorescens SBW25, and Xanthomonas campestris pv. vesicatoria strain 85-10, respectively (Fig. 3). The nucleotide sequence of EglC was deposited in the GenBank database (GenBank accession no. KT313000).

Figure 1 Amplification of a DNA fragment encoding the EglC gene from Citrobacter farmeri A1.

Lane M: DNA marker (100–2,000 bp); lane 1: 1,107 bp PCR product.

Expression and purification of EglC22b

The mature EglC gene without the signal peptide was cloned into the pET22b vector for expression. SDS-PAGE revealed that the apparent molecular mass of EglC22b was approximately 42 kDa (Fig. 4). This band was not present in non-transformed strains. The activity of the E. coli pET22b-EglC crude extract was 9.5 U/ml.

The crude EglC22b was purified using Ni2+-NTA affinity chromatography, as a protein with the expected MW (42 kDa) was present in the SDS-PAGE gel (Fig. 4, lane 5). The specific activity of the purified enzyme was 8.7 U/mg.

Figure 2 Conserved domains prediction of the EglC.

Conserved domains prediction was performed by NCBI CD-search software (https://www.ncbi.nlm.nih.gov/Structure/cdd/wrpsb.cgi).

Figure 3 Comparison of EglC protein sequences from different microorganisms.

(A) Citrobacter farmeri A1 (This study); (B) Escherichia coli CFT073 (GenBank accession no. AAN82779); (C) Pseudomonas fluorescens SBW25 (GenBank accession no. WP_012721724.1); (D) Xanthomonas campestris pv. vesicatoria str. 85-10 (GenBank accession no. WP_011348564.1); (E) Burkholderia sp. CCGE1002 (GenBank accession no. WP_012355143.1); (F) Cupriavidus taiwanensis (GenBank accession no. WP_012355143.1). Multiple sequence alignment was performed by Clustal Omega software (http://www.ebi.ac.uk/Tools/msa/clustalo/). Asterisks show residues in the column are identical in all sequences in the alignment. Dots show semi-conserved substitutions observed in the alignment. Colons show conserved substitutions.

Figure 4 SDS-PAG analysis of the recombinant EglC22b stained with Coomassie blue.

Lane M: protein MW marker (18.9–94.4 kDa); Lane 1: IPTG-induced E. coli pET22b-EglC; Lane 2: E. coli pET22b-EglC; Lane 3: IPTG-induced E. coli pET22b; Lane 4: E. coli pET22b; Lane 5: purified EglC22b.

Effects of pH and temperature on the activity of EglC22b

The activity and stability of EglC22b were assayed in CMC-Na at different pH values and temperatures. EglC22b showed optimal activity at pH 7.0 and exhibited >94% and >85% relative activity at pH 6.5 and 8.0 (Fig. 5A). EglC22b was highly stable at pH 3.5–7.5 and retained >70 % residual activity after 30 min of incubation in these buffers (Fig. 5B).

Figure 5 The pH properties of EglC22b.

(A) Effect of pH on the activity of EglC22b; (B) The pH stability of EglC22b. The EglC22b activity which was detected at optimal pH and temperature was regarded as 100%. All measurements of the present study were carried out in duplicate. Data were presented as means with standard deviation (SD).

The recombinant enzyme had an optimal temperature of 30–40 °C and exhibited >50% relative activity even at 5 °C (Fig. 6A). The activity of EglC22b was lost rapidly at temperatures higher than 60 °C (Fig. 6A). The thermotolerance analysis showed that approximately 90% of enzymatic activity was lost after 30 min of incubation at 60 °C (Fig. 6B)

Figure 6 The temperature properties of EglC22b.

(A) Effect of temperature on the activity of EglC22b; (B) The temperature stability of EglC22b. The EglC22b activity which was detected at optimal pH and temperature was regarded as 100%. All measurements of the present study were carried out in duplicate. Data were presented as means with standard deviation (SD).

Effects of chemical reagents and metal ions on the activity of EglC22b

The effects of various chemical reagents and metal ions on the activity of EglC22b were tested (Fig. 7). The presence of Cd2+, Zn2+, Li+, Triton X-100, DMSO, acetonitrile, Tween 80, EDTA, and SDS inhibited the activity of EglC22b. However, the presence of Co2+ and Fe3+ increased the activity of EglC22b.

Figure 7 Effects of chemical reagents and metal ions on the activity of EglC22b.

The EglC22b activity which was detected at optimal pH and temperature was regarded as 100% (Control). All measurements of the present study were carried out in duplicate. Data were presented as means with standard deviation (SD).

Discussion

Our previous study showed that C. farmeri A1 from the gastrointestinal tract of R. labralis is an effective cellulase-producing bacterium (carbon source: CMC-Na; X Bai & T Shao, 2015, unpublished data). Although many endoglucanases from various microorganisms including fungus, bacteria, and actinomycetes have been studied, no endoglucanase from C. farmeri has been characterized to date (Shelomi et al., 2014; Asgher, Ahmad & Iqbal, 2013). In the present study, a cold-adapted endoglucanase was cloned from C. farmeri A1 and expressed in E. coli. The recombinant enzyme was purified to determine its biochemical properties.

The EglC gene was obtained from the DNA of C. farmeri A1 by TAIL-PCR. The deduced amino acid sequence of EglC is highly similar to the amino acid sequence of the EglC from E. coli CFT073. BLAST analysis suggested that EglC is a member of the GH 8 family. The amino acid sequence of EglC included a signal peptide (21 amino acids) and mature protein. For characterization, the mature protein of EglC was expressed in E. coli. SDS-PAGE revealed that the molecular weight (MW) of EglC22b was approximately 42 kDa, which was similar to the theoretical MW of 39.1 kDa.

EglC22b was characterized as a neutral enzyme (i.e., active at neutral pH). The recombinant EglC22b was active at pH 6.5–8.0 with optimum activity at pH 7.0, which was similar to the characteristics of C. farmeri EglC and the endoglucanase (Umcel9B) isolated from compost soil microorganisms (Pang et al., 2009). The present results indicate that EglC22b was highly stable at pH 3.5–6.5 for 30 min. Similar pH stability was also observed in the EG5C endoglucanase from Paenibacillus sp. IHB B 3084 (Dhar et al., 2015).

Although many endoglucanases have been studied, only few cold-adapted enzymes have been reported (Fu et al., 2010; Ueda et al., 2014). It is known that cold-adapted enzymes show relatively high activity at low temperatures and have a low optimal temperature and poor thermal stability (Dhar et al., 2015). EglC22b showed optimal activity at 30–40°C and more than 50% maximal activity at 10 °C. It also had relatively poor thermal stability, where approximately 90% of its activity was lost after incubation at 60 °C for 30 min. These temperature properties suggest that EglC22b has typical characteristics of cold-active endoglucanases, which was identical to the EglC of C. farmeri A1. Similar results were also previously observed for endoglucanase and cellulase from Paenibacillus sp. IHB B 3084 (a psychrophilic deep-sea bacterium) and Eisenia fetida (Dhar et al., 2015; Yang & Dang, 2011; Zeng, Xiong & Wen, 2006; Ueda et al., 2014). In contrast, the mesophilic and thermmophilus endoglucanases were previously shown to rapidly lose activity at temperatures below 20 °C (Bischoff, Liu & Hughes, 2007; Li et al., 2011)

Several reports have indicated that Co2+ can enhance the activity of endoglucanases from E. coli Rosetta 2 and Aspergillus niger (Rawat et al., 2015; Martin et al., 2014). In the present study, the activity of EglC22b was also increased by the presence of Co2+. Furthermore, SDS and DMSO were found to almost totally inhibit the activity of EglC22b (<10%), which was in agreement with a previous study (Dhar et al., 2015; Manavalan et al., 2015).

Conclusion

An EglC gene was cloned from C. farmeri A1 and then expressed in E. coli. Biochemical characteristics of EglC22b indicated that it was a low-temperature-active endoglucanase. Cold-adapted endo-1,4-β-glucanases can protect thermolabile substrates, reduce energy consumption, and minimize the rate of nonspecific chemical reactions. Further studies in our laboratory will focus on its applications in the feed, food and silage industry

Supplemental Information

Supplemental Information 1 Effects of pH on the activity of EglC22b

Click here for additional data file.

Supplemental Information 2 pH stability

Click here for additional data file.

Supplemental Information 3 Effects of temperature on the activity of EglC22b

Click here for additional data file.

Supplemental Information 4 Temperature stability

Click here for additional data file.

Supplemental Information 5 Effects of metal ions and chemical reagents on the activity of elgC22b

Click here for additional data file.

Additional Information and Declarations

Competing Interests

Author Contributions

DNA Deposition

Data Availability

The authors declare there are no competing interests.

Xi Bai conceived and designed the experiments, performed the experiments, analyzed the data, contributed reagents/materials/analysis tools, wrote the paper, prepared figures and/or tables, reviewed drafts of the paper.

Xianjun Yuan conceived and designed the experiments.

Aiyou Wen and Yunfeng Bai analyzed the data.

Junfeng Li performed the experiments.

Tao Shao conceived and designed the experiments, wrote the paper, reviewed drafts of the paper.

The following information was supplied regarding the deposition of DNA sequences:

GenBank accession numbers: KT313000.

The following information was supplied regarding data availability:

The raw data has been supplied as a Supplementary File.

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
