# Peer review of "Cloning, expression and characterization of a cold-adapted endo-1, 4-β-glucanase from Citrobacter farmeri A1, a symbiotic bacterium of Reticulitermes labralis"

_PeerJ, doi:10.7717/peerj.2679_

## Round 0.1 · original submission · Major Revisions

· Academic Editor

Major Revisions

Although reviewer 2 has relatively few (but significant) remarks, I share reviewer 1's opinion that the manuscript needs substantial work to be considered for publication.

I encourage the Authors to re-assess clarity and accuracy of the experimental section in general, starting from (but not necessarily limiting to) the reviewers' remarks.

I will be looking forward to receiving a revised version of the manuscript.

Reviewer 1 ·

Basic reporting

The manuscript entitled “Cloning, expression and characterization of a cold-adapted endo-1, 4-β-glucanase from Citrobacter farmeri A1, a symbiotic bacterium of Reticulitermes labralis” addresses the recombinant production of a bacterial endoglucanase and the analyses of its biochemical properties. The Authors describe thermal and pH inactivation, and the response to various metal ions. The persistence of >50% relative activity at 5 °C make the Authors to conclude that EglC is a cold-adapted enzyme.

The English of the article does not conform to professional standards. Several unclear points are exposed in the “Comments for the Authors” section.
The Introduction fails to highlight the importance of cold-adapted endoglucanases in biotech processes (in which field cold-adaptation is desirable? Degradation of lignocellulosic material for biofuel production does not appear the most relevant field). The references to a previous, stil unpublished work from the same Authors make insufficiently clear some part of the manuscript.
Some figures are not appropriately described and labeled (see in the “Comments for the Authors” section).

Experimental design

The cloning and subcloning procedure is not clear.
The purity degree of the enzyme should be expressed in terms of specific activity (U/mg), especially in view of a comparison with different enzyme preparations.
See details in the “Comments for the Authors” section.

Validity of the findings

No comments

Comments for the author

I've copied some sentences of the manuscript (numbers of rows included) and reported below my comments

59 "The psychrophilic
60 proteases have the merit of energy savings in the lignocellulose degradation process for further
61 production of energy fuels.

To my knowledge, the process of lignocellulose degradation most often includes a steam explosion step. The subsequent steps of enzymatic degradation require the cooling of the mass to avoid thermal denaturation of enzymes. Moreover, warming occurs during sugar fermentation. Overall, thermozymes are considered more suitable catalysts in the lignocellulose degradation for production of biofuels. Which kind of lignocellulose degradation process would employ a cold-adapted enzyme?

68 In our previous study, a
69 endoglucanase-degraded bacterium, Citrobacter farmeri A1, was isolated from...

I guess the Authors intended an “endoglucanase-degrading bacterium”

111 The recombinant plasmid pET22b-EglC was separately transformed into competent cells(E. coli
112 BL21) and plated on LB agar containing 100 μg·ml-1 ampicillin to select positive transformants.
113 The positive clones were checked by PCR and sequencing

The cloning and subcloning procedure is not clear. Do the Authors sequenced 1) PCR-amplified DNA from pET21 positive clones or 2) in vivo amplified DNA from DH5alfa cells? What means “plasmid pET22b-EglC was separately transformed”?

170 The activity of recombinant E. coli pET22b-EglC was 9.5 U/ml.

Do the Authors refer to the activity of E. coli crude extract? Then, which is the specific activity of purified enzyme (usually in U/mg)?

171 The crude EglC22b was purified using Ni2+-NTA affinity chromatography, as there was only
172 one band with a molecular mass of approximately 42 kDa in the SDS-PAGE gel (Fig. 3).

The lane of IPTG-induced crude extract is rather “crowded”. I would not say that there is only one band of 42 kDa. A relevant point is that a protein of expected MW appears upon IPTG-induction.

195 The EglC gene was obtained from the DNA of C. farmeri A1 by TALL-PCR. The deduced.

Correct TAIL-PCR

205 glycosyl hydrolase family 9 (Fu et al., 2010).The recombinant endoglucanase expressed in E.
206 coli, pET22b-EglC, showed higher activity (9.5 U/ml) than the original wild-type endoglucanase
207 from C. farmeri A1 (0.8 U/ml) and the recombinant endoglucanase expressed in E. coli EF-EG2
208 (1 U/ml) (Ueda et al., 2014)

Which are the bases for such a comparison? Different degrees of purity for recombinant and natural EglC can obviously account for their different activity. Why the Authors do compare recombinant EglC with EF-EG2?

230 Furthermore, methanol,
231 DMSO, and SDS were found to almost totally inhibit the activity of the enzyme, which was in
232 agreement with a previous study (Dhar et al., 2105; Manavalan et al., 2015). The reason may be
233 that these chemical reagents could destroy the protein structure, leading to enzyme degeneration.

That methanol, DMSO, and SDS may impair the activity and even the structural integrity of EglC does not seem a peculiar trait. And also the explanation appears too generic. It may deserve some interest the kinetics of inactivation upon incubation with these common denaturing agents.

Legend of Figure 2: asterisks, dots and colons below aligned sequences are not explained.

Figure 3: The title of figure 3 (“Results of SDS-PAGE”) is poorly informative.

Figure 4: The title should be reformulated. Moreover, the legend refers to “enzymes” or “endoglucanses”. Why the use of plural nouns?

Bibliography: The Authors of first cited article (Muhammad Asgher, Zanib Ahmad, Hafiz Muhammad Nasir Iqbal) are missing!

Several time the Authors refer the use of the compound “Trion X-100”. I guess they mean “Triton X-100”.

Reviewer 2 ·

Basic reporting

No comments

Experimental design

Please check the following points.

1. Line 60: Please rewrote from "The psychrophilic protease" to "The psychrophilic cellulase?

2. Line 69: please rewrote from "endoglucanase-degraded" to "endoglucan-degraded?

3. You should be showed specific activity toward CMC and other cellulose substrates (soluble and insoluble cellulose).

4. You should be checked the hydrolysis products of cellulose and/or cello-oligosaccharides. You can also identified whether this enzyme is endo or exo typed cellulase from this experiment.

5. Fig. 4B: Please rewrote from " Relative activity (%)" to "Residual activity (%)".
You should be identified the residual activity under pH 3.5 and over pH 6.5.

6. Fig 5B: Please rewrote from " Relative activity (%)" to "Residual activity (%)".
You had better check the 20oC, and 30oC of residual activities.

Validity of the findings

No comments

Comments for the author

This paper is described about "Cloning , expression and characterization of a cold-adapted endo-1,4-beta-glucanase from Citrobacter farmeri A1, a symbiotic bacterium of Reticulitermes labralis." This paper is sounds good. But, I have some comments and recommendation.

---

## Round 0.2 · Minor Revisions

· Academic Editor

Minor Revisions

Although the Authors addressed most concerns raised by the previous reviewers, I believe further work is required for this paper to be acceptable for publication. In most cases, it is a matter of improving clarity and conciseness. Details below.

Experimental section
Repeats
The number of repeats conducted should be stated. Currently, error bars are displayed on plots, but the information about how many repeats were conducted, and which value is reported as error bar, should be described in the Methods section, and briefly referred to in figure legends.

Previous work
At the beginning of the Discussion (line 200), the authors refer to their previous work that led to the identification of cellulase-producing bacteria. They do not, however, discuss the question of whether the properties of EglC match those of the enzyme previously purified from the natural host, as a different enzyme could have been responsible for the activity detected from C. farmeri. If experimental data on the previously isolated enzymes cannot be provided, a discussion of the differences (or similarities) between natural and recombinant enzymes would be welcomed.

Figure 2
The Authors refer to this figure in line 168 to support the statement about length and MW of the isolated proteins. As Fig 2 only contains a multiple alignment, hence this reference looks inappropriate and should be removed.
Moreover, in line 171 the Authors say that from the multiple alignment reported in Fig 2 it was possible to conclude that the isolated protein belongs to glycoside hydrolase family 8. How was this determined? Is the alignment with 5 other protein sequences enough to assess which class EglC belongs to? Additional details should be featured in the figure legend, or in Methods.

Figure 5
Why wasn't stability tested at lower temperatures? If the point of the manuscript is reporting on EglC's cold-adaptation, one would expect to see if the enzyme is significantly more stable under these conditions.

Figure 6
The condition used as 100% is not shown on the plot, or stated in the legend. The ions names in the chart x-axis should be converted to the appropriate superscript format. Finally, it would be useful if the Authors were to explain (in the main text) why organic solvents (DMSO, acetonitrile) and denaturants (Triton X-100, Tween 80 and SDS) were tested.


Text edits
Although the level of written English has improved significantly, some typos need correcting still. Examples are:
line 56: "non-thermal stability"
line 58: "and" should be added after the last comma
line 110: "was" should be replaced by "were"
line 113: "svector" should read "vector"
line 240: "was" instead of "were"
legend to figure 1: "fragmentsencoding"
The Abstract, in particular, stands out as not having been corrected to the same standards. Revision is advisable.

Lines 78-79: The Authors mention how this is the first report of a similar enzyme from their specific source, This is a rather restricted goal, and I do not believe it's a main achievement in this report. It doesn't look appropriate as closing statement of the introduction.

Lines 214-217: the discussion of the MW of few other enzymes adds very little to the validity of the Authors' findings. If such comparison is essential, it should be mentioned either more superficially, or in further detail (e.g. by adding a table comparing a larger number of enzymes). To be honest though, small variation in length are the norm in enzymes from secondary metabolism, so I am not sure of which point the Authors want to make from this comparison.

Lines 229-231 The discussion on the possible reasons for the low thermal stability of EglC is far too speculative and superficial to be acceptable. Not only there are at lest 2 cysteine residues that could form disulfide bridges, the molecular determinants for structural stability extend to far more complex features. Since the point of the manuscript is not to explain the lack of thermal stability, I believe this sentence should be removed.

Lines 241-244 Similarly, the discussion of the effects of SDS and DMSO is a speculation based on trivial biochemistry classifications (SDS is a denaturant, and DMSO a chaotropic agent) that does not add any information, and should be removed.

Lines 251-253 Although future work can be referred to in a manuscript, this sentence is very specific on relatively marginal improvement. Describing what the authors want to achieve ultimately by the characterisation and implementation of ElgC in biotechnology application would be a more valuable information to conclude the discussion on.

Figure 5. It seems that a working draft legend has remained in Panel B. Please remove it.


I will be looking forward to receiving a revised version of the manuscript.

Reviewer 2 ·

Basic reporting

The revised manuscript is rewritten according to the reviewer's comments. I think that this paper might be accepted to Peer J.

Experimental design

The revised manuscript has a good experimenetal design.

Validity of the findings

The revised manuscript has a good validity.

---

## Round 0.3 · accepted · Accept

· Academic Editor

Accept

The manuscript has been revised by the Authors, and is now suitable for publication.